# Exploring Managerial Job Demands and Resources in Transition to Distance Management: A Qualitative Danish Case Study

**DOI:** 10.3390/ijerph20010069

**Published:** 2022-12-21

**Authors:** Nelda Andersone, Giulia Nardelli, Christine Ipsen, Kasper Edwards

**Affiliations:** 1DTU Management Department, Technical University of Denmark, 2800 Kgs. Lyngby, Denmark; 2Independent Researcher, 2100 Copenhagen, Denmark

**Keywords:** manager well-being, distance management, job demands, job resources, first-line management, COVID-19

## Abstract

Organizations worldwide have shifted to working from home, requiring managers to engage in distance management using information and communication technologies (ICT). Studies show that managers experience high job demands and inadequate guidance during COVID-19; therefore, the transition to distance management raises questions about the increase in managerial job demands and the impact on managers’ well-being. This study aims to explore first-line managers’ perceptions of job demands and available resources during the first year of the pandemic and understand the implications for first-line managers’ well-being. First-line managers face complex and conflicting demands, making them more challenged in their management task than other management levels. We used the job demands–resources model in this qualitative, longitudinal empirical study. The study draws on 49 semi-structured interviews with seven first-line managers from a large pharmaceutical company in Denmark, whom we followed throughout the first year of the COVID-19 pandemic, from May 2020 to May 2021. Our findings suggest that the first-line managers perceived increased emotional and practical demands. While the managers appreciated the initial guidance provided by the organization, they perceived the organizational support as outdated and superficial. As a result, to cope with the uncertainty caused by the pandemic and the shift to distance management, the managers relied on work engagement enablers such as social support. Even though the COVID-19 pandemic portrays unique circumstances in transitioning to distance management that require further exploration outside the COVID-19 context, the insights from this study can assist organizations in developing awareness about transitions to better support first-line management to embrace changes in the future.

## 1. Introduction

The coronavirus (COVID-19) pandemic affected organizations worldwide, making them move most of the workforce from physical workspaces to home offices. In response to the pandemic experiences, many companies have been changing their work organization and exploring remote work, e.g., working from home, even though most organizations were not prepared to support such a practice [1,2]. Studies indicate an increase in the volume of development and production of ICTs goods and services since 2020, thus reflecting the significant increase in adopting remote work [3].

Before COVID-19, companies worldwide have been embracing remote work including working from home. The Sixth European Working Conditions Survey [4] shows that 37% of the Danish workforce had engaged in working from home, and when compared to other European countries, the Danish workforce was highly experienced in working from home before the pandemic. Nevertheless, internationally, most workers had limited experience and knowledge of working remotely in a collective way. Hence, this transition turned into an experiment of remote working [2] and distance management.

The changes included managers engaging in a new managerial task where they shifted to managing their employees from a distance. The use of ICT in conducting work was widely adopted and used before the pandemic [5], enabling workers to access work from outside of workplaces at any time [6]. COVID-19 further enhanced this adoption, and managers needed to rely on ICT to fulfill their managerial duties.

In this paper, we explore first-line managers’ perceptions of job demands and available resources to them when transitioning to distance management during the first year of the pandemic. Based on our findings on their perceived job demands and resources, we draw the implications on the first-line managers’ well-being.

Most empirical studies exploring job demands and resources tend to focus on employee working conditions (e.g., [7,8,9,10]). Studies on experiences during COVID-19 predominantly cover employee work conditions (e.g., Jamal et al., 2021; Meyer et al., 2021; Lilja et al., 2022) and surprisingly little attention has been placed on the experiences and the well-being of those leading employees during COVID-19 [11].

However, it is vital to study managers’ experiences as managers’ well-being influences how they lead, the quality of their work, and the relationships with their employees [12]. When a manager experiences good well-being, they tend to adopt healthier management practices that may promote employee well-being [13,14]. Instead, managers experiencing stress can potentially compromise their healthy management practices and negatively affect how they lead people [14]. Hence, managers’ behaviors and leadership influence employee well-being as well [14,15,16,17]. Therefore, there is a need to further the understanding of managerial working conditions. To expand on the managerial working conditions, we use the following research question to guide our study: How do first-line managers perceive their job demands and resources when transitioning to distance management during the COVID-19 pandemic? This study contributes to the managerial well-being literature by outlining the constituents of emotional and practical job demands, and work engagement enablers, and demonstrates the influence they have on managers’ job quality in the transition to distance management. We also extend the emerging research on work during the pandemic that has primarily focused on employee experiences [11].

The remainder of this paper is distributed in the following way. First, we introduce our theoretical background. Then we explain our methodological approach to approach this study. Further, we present our analysis, and then discuss the findings within the theory and outline the limitations of this study. We conclude the paper by formulating the implications for the management practice and considerations for further research.

### 1.1. Distance Management during the COVID-19 Pandemic

Managing employees from a distance is different from managing individuals located at the same location [18]. Generally, distance management refers to managing the workforce from various locations within and outside the company, home offices, client sites, and other places than the traditional workplace [19]. Time, space, and culture are the components establishing the distance between the managers and employees [20]. Instead, the use of ICT creates connectivity [21]. However, research has been focusing on how to lead and bridge work and people in virtual teams i.e., non-co-located teams (e.g., [22,23]). Through the reliance on ICT, on the one hand, organizations have become more productive [24] and cost-efficient [25]. On the individual level, workers face better control over organizing their work and experience a better work–life balance due to reduced commute time [19]. However, working from a distance also promotes a loss of sense of belonging, lessening knowledge sharing, and social interaction, and because people’s work behaviors become less visible [26,27], having a distance between managers and employees shifts the nature of managerial control [21,28]. As a result, managers perceive obstacles in overseeing employees’ work [29]. Previous studies suggest that the shift to distance work for managers tends to present significant demands, thus extending their work [30].

Because of the extensive reach and significant influence of the pandemic, the pandemic presented unique demands meaning that the previously accumulated knowledge could not apply to the COVID-19 context [2,31], especially because studies tend to focus on virtual teams where individuals rarely collaborate face-to-face or where remote work is practiced occasionally by some organizational members [2]. The COVID-19 pandemic also introduced many managers and organizations to distance management with no collective experience to draw on. 

During the pandemic, managers experienced management from a distance as challenging [19,20]. Giusino et al. [32] observed an increase in managers’ workload during COVID-19, raising time pressure and prolonging their work hours, especially affecting lower-level managers as they were challenged in needing to respond to employees while remaining on track with achieving organizational goals [32]. However, if the managers are demanded to work more deliberately when leading from a distance, this may affect managers’ well-being negatively and impact the development of potential burnout [33]. In this light, it becomes essential to understand the job demands and resources for managers during COVID-19.

### 1.2. First-Line Management

Management positions tend to be assumed to be appealing due to recognition, rewards, and the power to influence the organization, experience autonomy, and participate in decision making [13]. While managers tend to experience more control, influence, and autonomy and overall have access to more resources than employees [34], managerial work can be highly challenging because of overall stressful working conditions [12,35,36]. Managers face high job demands [34,37] that result in work overload, ambiguous roles, working beyond official work hours, and work–life conflict [34,35].

Previous studies have indicated that first-line managers tend to face complex and conflicting demands and, thus, are more prone to experience stress at work [34,35] even more than other management levels [38]. First-line management is responsible for managing daily work operations, and their core responsibilities include short-term focus, immediate direction, and performance-oriented supervision of their employees within a narrow and bounded unit [39] requiring their presence at work every day [34]. 

Over the last decades, first-line managers have faced changes in the content and volume of their jobs, i.e., their role has expanded, and first-line management has been supplemented with responsibilities that previously belonged to middle management (e.g., business management responsibilities), blurring the middle and first-line management roles. As a result, declining the quality of their work [40]. First-line management’s declining work quality indicates that the first-line managers may not be able to bear their additional job responsibilities [39].

However, first-line managers face challenging work because they are accountable for work effectiveness yet have a low influence over the decisions from higher management levels that could determine effectiveness. Simultaneously, they hold low control over things for which they are kept accountable [39]. They receive information about organizational changes decided by senior management, which they need to comprehend and implement, dealing with consequences experienced by employees such as anxiety and confusion [39,41]. First-line managers’ low influence and involvement in decision making and the lack of adequate management training to fulfill their additional responsibilities result in them struggling with their job demands [39]. Hence, line managers tend to be more vulnerable to experiencing reduced well-being than higher levels of managers [15,34].

Studies indicate that the shift to distance management was challenging, presenting new demands for managers [30], first-line management being an especially vulnerable group more prone to dealing with stress [34,35] than higher managerial levels [38]. Hence, it makes it imperative to observe the job demands and resources faced by the first-line management in the COVID-19 context.

### 1.3. Job Demands and Resources

Any line of work contains specific job demands and job resources that boost motivation and engagement in work as well as influence the development of work-related stress [42,43,44]. Managing employees through changes following COVID-19 can be especially demanding and complex due to the experienced uncertainty [45], thus affecting managers’ work activities by causing more pressure [13].

The job demands–resources theoretical framework assumes the interplay of two diverse categories of job characteristics (i.e., job demands and job resources) and associates them with experiencing well-being at work (e.g., satisfaction with work, and work engagement) [46]. The theory allows gaining awareness about the work environment’s influence on an individual’s well-being and performance [47]. 

Job demands refer to psychological, physical, social, and organizational elements that require demanding physical, emotional, and mental input from individuals [48,49]. For example, increased workload, time pressures, fewer breaks, role ambiguity, role conflicts, interpersonal conflicts, job insecurity, and any other requirements that demand continuous physical, emotional, and mental input classify as job demands. Job demands tend to be associated with activation leading to stress and draining of individuals’ mental and physical resources over time [43]. For example, facing extensive emotional demands and work overload for a prolonged time tends to contribute to anxiety and deplete an individual’s mental and physical resources leading to exhaustion and other health problems [43]. As a result, having high job demands has a negative connotation. The theory assumes that job demands reduce well-being whereas having sufficient resources tends to reduce the impact of work demands [50], even though the resources do not directly deal with job demands [35,42]. 

Job resources relate to social and organizational job aspects as well as personal resources (e.g., physical, psychological). These resources alleviate reaching work goals, help individuals cope with job demands, stimulate learning and development [42,43], improve well-being [46,48,51], and lead to positive organizational outcomes [52]. Therefore, resources are vital in attaining work goals [15]. These aspects include wages, a supportive work environment with a sense of solidarity with others, autonomy, career opportunities, job control, knowledge, training, task significance, feedback, role clarity, self-efficacy, and having organization-based self-esteem (i.e., belief about own value and perceiving being valued at work)^42^. Job resources such as social support and helping others, team cohesion, training, and communication are considered the most effective support strategies and are highly influential when coping with high demands [42]. Having high job resources is especially important in conditions of high job demands as resources allow for maintaining work engagement and coping with the demands [43]. Facing significant demands with limited resources, i.e., where demands require effort that is higher than the individuals’ capacity for achieving work goals, leads to the imbalance becoming a physical and psychological stressor, especially when insufficient recovery is involved [47,53]. This gradually leads to irritability, cynicism, disengagement, and burnout, i.e., emotional and physical exhaustion [50,54]. This may contribute to plummeting work performance and, eventually develop adverse organizational outcomes [52].

## 2. Materials and Methods

The study uses qualitative semi-structured interviews with seven first-line managers as a qualitative research method.

### 2.1. Case Company

We conducted a longitudinal in-depth exploratory case study in a large Danish pharmaceutical company from May 2020 to May 2021, during the first year of the COVID-19 pandemic, thus following the consequent lockdowns and re-openings of the workplaces. For example, from March 2020 until May 2021, Danish workplaces experienced five subsequent states: (1) lockdown and the workforce working from home to (2) first re-opening following the lockdown, (3) new restrictions announced by the Danish government, (4) the second lockdown, and (5) the re-opening of the workplaces and society following the second lockdown [55]. We accessed the company when the company was engaging in the first opening after the first lockdown.

We specifically followed and explored the distance management experiences of seven first-line managers as they alternated between onsite and distance management. Their reports were knowledge workers, and both the reports and the first-line managers predominantly were located in Denmark. 

To select the case company, we followed a criterion sampling approach [56], developing pre-defined elements of interest before fixating on a specific case. Our selection criteria focused on the size of the company, e.g., large with a workforce comprising 250+ employees, location e.g., in Denmark, domain, e.g., knowledge-intensive organization, and included complementary features such as that the company was transitioning to and engaging in remote work and distance management. Furthermore, we were interested in engaging with a company that values employee well-being. Therefore, we selected a case company that emphasizes employee well-being and prioritizes ensuring its workforce with good working conditions. We used Glassdoor (i.e., www.glassdoor.dk, a website collecting anonymous company reviews from current and former employees) to confirm this. Lastly, a significant precondition was that the potential case company would permit our access to them for at least six months. 

### 2.2. Procedure

We drew 10 rounds of semi-structured interviews with seven first-line managers that we conducted every 4–6 weeks. Conducting qualitative semi-structured interviews allowed us to obtain rich and diverse data (Myers & Newman, 2007) that explored the invisible aspects of people’s outer and inner worlds and clarified the interpretations of retrospective and real-time reflections of their life experiences [57].

Each interview round comprised 8 to 10 individual interviews, resulting in 49 interviews. Over the 10 interview rounds, we adhered to the same seven informants; see Table 1 for grasping our informant participation in the interview rounds. We conducted the interviews in English. Each interview lasted between 15 and 40 min. Participation in the interviews was voluntary. The interviews were conducted in Microsoft Teams (MS Teams) and afterward transcribed verbatim using Otter.ai transcription service and double-checked for the accuracy of transcripts. When editing the transcripts, we removed the informant names, instead replacing them with an informant number, in that way anonymizing our informants. In total, four interviewers were involved in the data collection, and the interviews were conducted in a rotating manner so each interviewer would have the possibility to obtain insight into each informant’s situation. Table 1 shows informant participation in the interviews according to the chronological developments following the pandemic in Denmark. After round 4, three of the informants withdrew their participation. Grey fields in Table 1 indicate a missing interview.

To conduct our interviews, we followed an interview guide. Our interview guide focused on three areas: (1) reflections surrounding ICT use and meeting frequency; (2) reflections on distance management, well-being and performance, and team environment; (3) theme exploration. In each interview round, we explored a new theme. The themes for each round emerged through our gained insights. The theme exploration included areas such as trust, managerial expectations in remote work, and changes in the managerial role. For rounds one to three, we used the initial interview guide (Section A.1.); however, from rounds four to ten, we used a revised interview guide (Section A.2.) due to the shift to hybrid work, e.g., navigating work between onsite and offsite, following the withdrawal of COVID-19 restrictions in Denmark. Our revised interview guide provided a structure to capture the shifts in work location, uncover managers’ actions towards employees, and exploring their overall experiences managing from a distance.

### 2.3. Participants

The first-line managers involved in this study headed between nine and twenty-two direct reports. The first-line managers had some distance management experience with leading one to four employees who were working abroad (e.g., Informants 2, 3, 4, and 7); however, only one of the managers (Informant 5) had led a virtual team having employees work from multiple locations. Informants 1 and 6 were completely new to the distance management task. Overall, before the outbreak of COVID-19, working remotely was not the standard at the company, and the core of the teams was in Denmark. The standard was that both employees and managers worked one to two days a month from home whenever they had to juggle private appointments. The remainder of the time, the company’s workforce operated from the company’s premises. Hence, only one of the informants (Informant 5) could draw on experience leading a whole team from a distance. Table 2 demonstrates an overview of the informants’ experience in a management role and at the distance management task. The grey fields in Table 2 indicate that these informants had not engaged in distance management tasks before the pandemic.

### 2.4. Data Analysis

The interview transcripts were our primary source of data, and we analyzed our data using the qualitative data analysis software Atlas.ti. The main author (N.A) performed multiple levels of coding. Each coding round was assisted and reviewed by another researcher (G.N). The development of distinct categories, themes, and aggregate dimensions was discussed, reviewed, and agreed upon between all the authors. 

We followed the Gioia method [58] to systematically approach the analysis and develop a data structure. Our exploration was guided by our research question to explore the first-line managers’ perceptions of their job demands and resources during the shift to distance management.

The analysis process began by performing a first-order analysis where we read each interview transcript. When engaging with the transcripts, we sought to identify consistent patterns emerging from our informants’ descriptions of their experience across the corpus of the data. This was carried out by taking each interview transcript and marking sections of informant expressions that offered relevance to the research question. We assigned a dedicated label, i.e., a code to the sections we found relevant. In the first-order analysis, we identified 30 consistent codes and marked the data segments in the whole corpus of data according to these identified codes.

In the following step, i.e., second-order analysis, we focused on the initial 30 codes and by outlining each code, we sought similarities with other codes. By distilling and merging the codes, we organized the codes in overarching categories. We labelled the categories by linking the prior identified informant terms to our theoretic insights and provided a short description to each category to guide us when performing the next coding round. We then revisited our interview transcripts, and analyzed the data by distilling the corpus of data into eight distinct, overarching blocks of data, i.e., according to our identified categories (as shown in Appendix C, Table A1). 

Consequently, by linking our insights with the theoretical concepts in the existing literature, we grouped the eight categories further into three aggregate dimensions. While the themes within each aggregate dimension share similarities, they represent a different angle of the aggregate dimension. See Appendix B, Figure A1 for the data structure and Appendix C, Table A1 for a representation of our progression from the data to codes, categories, and aggregate dimensions in the analysis.

## 3. Results

In this section, we present the first-line managers’ perceptions of their job demands and resources during the first year of the COVID-19 pandemic. We provide detailed presentations of the prominent themes that were expressed by all first-line managers interviewed. The headlines in this section represent the themes we identified via the analysis. First, we demonstrate the perceived demands (e.g., emotional and practical demands) that we identified and outline the sub-categories that constitute each demand. We also elaborate on the perceived resources (e.g., work engagement enablers) that we identified and elaborate on the related sub-categories. This representation lays the foundation for the later discussion, where we elaborate on the implications for the first-line managers’ well-being.

### 3.1. Emotional Demands

Emotional demands include those demands that require facing, tolerating, and responding to a high degree of uncertainty and provision of significant care and support for others. 

#### 3.1.1. Dealing with Uncertainties

Due to the spread of COVID-19, employees and managers left the physical office and during this time, they needed to alternate between managing in the office and from home. First, the organization instructed the whole workforce to work in home offices and then brought employees back to the offices. Afterward, they again transitioned into hybrid work, and shortly after, requested employees to return to home offices due to the Danish government’s announcement of the second lockdown. During the second lockdown, the organization permitted office attendance to be up to 50% in each team, later reduced it to 25% of attendance, and then increased it to 30% for business-critical tasks. This ongoing change and the need to adapt to back and forth work situations caused confusion and drained first-line managers:


*“It’s becoming increasingly frustrating. Now that fall is coming, people tend to get infections…it is just getting increasingly difficult figuring out who needs to stay home when, and who needs to get tested…everyday people are checking in: “I have a sore throat, what should I do?” It is a lot of work for me to make decisions. Also, it seems that daily, we need to switch from physical to virtual and the other way around” (Informant 1)*


All the first-line managers preferred managing their teams from the physical workplace because they found that seeing people face-to-face regularly was important in their managerial work. When working at the office, the first-line managers tended to naturally observe employees in a day-to-day setting, acquiring a sense of how their employees were dealing with their tasks. The first-line managers claimed that seeing people in the physical workplace provided them with insight through which they knew how people were doing. Recording and decoding employee behaviors were inherent in the workplace; however, when facing the pandemic and working from home, there was no possibility for the first-line managers to observe their employees in the same way:


*“I always have this underlying concern about people: “How are they doing?” I am not 100% sure that they are at an optimum. Going to the office and seeing people every day is a different thing. You can read them…that is what I have realized much more than I had been conscious about, I tend to read people, their facial expressions, bodily posture, the way they walk, the way they drink their coffee, the way they sit at a meeting. That is taken away, that part I don’t have anymore” (Informant 3)*


Not seeing employees daily contributed to first-line managers doubting how employees were dealing with their tasks, and simultaneously, the managers feared that distance would detach employees from the team:


*“I feel that managing a team is just not the same. I like having a close connection with my team, knowing how they feel, and knowing that they are doing what they like to do. Also, the proximity and tightness of the team spirit is just weaning off a little bit as we cannot meet for anything” (Informant 1)*


Meanwhile, employees needed to figure out how to deal with commuting, new regulations, family responsibilities, and their home office setup. The first-line managers needed to continue delivering on organizational goals, as usual, guide employees, and support them emotionally and practically. Synchronously, they were adjusting to the continuous shifts throughout the pandemic, which required immediate decision making. The managers experienced the constant shifts as a struggle due to the high degree of uncertainty, and also they were unsure of the consequences of their decisions and actions. Concurrently, the managers were facing fears that they were losing track of their employees and that lack of regular face-to-face interactions would lead to a loss of team cohesion. 

#### 3.1.2. Prioritizing Employees

It was important for the first-line managers to know how well their employees were coping with the COVID-19 pandemic and were concerned about its influence on their workload and well-being. Knowing that employees were well and showing that they cared was ultimately how the managers defined caring in their managerial work. Showing care to employees was a way to develop and maintain trust with employees. They achieved this during the pandemic by dedicating specific time weekly for each employee, where they discussed each employee’s current work and life events. During the interactions, the managers focused on both verbal communication and subtext whenever facing employees, trying to picture *what is going on in their heads* and thus *tuning in* with the employees:


*“Maintaining close contact to direct reports—that’s crucial in my eyes to get a feeling for how they are coping. Some people…they may not necessarily tell initially, they will take a time to unravel what is going on. It is about taking the time, talking to people, getting a feeling for how they’re coping, listening carefully to what they’re saying, but also what they’re not saying—so applying listening skills” (Informant 5)*


The first-line managers emphasized that attentive listening is a way to show care. When interacting with employees, the managers demonstrated consideration for people, acknowledging the difficulties they experienced when working from home during the pandemic. They brought up stress in their conversations and gave room for employees to share their thoughts and reflections, encouraging employees to *air* their frustrations. Overall, repeatedly the managers brought up that they felt they needed to show up in a certain way, whenever facing their employees online. The managers perceived that they needed to portray themselves as available, present, and highly attuned leaders at all interactions. They informed employees of their availability, suggesting employees reach out via email or phone without the fear of disturbing:


*“I’m trying to be available. My calendar looks awful. I have meetings all the time. But I try to just contact them (employees). They’re also really good at just reaching out: “Can you call me when you have five minutes, I just want to talk about X” and then it’s up to me to get back to them” (Informant 4)*


By showing commitment and attending to employee needs, the managers felt they cared for their employees. We observed that these actions translated into first-line managers’ extending their work hours and that the additional demands and long hours in front of the screen negatively affected managers’ well-being:


*“In this period that we have been going through, it is important to show that I am committed, that I am interested in what they are doing. I think that has been even more important than it normally is. I found myself attached to my computer in meetings from early morning to the late evening, almost constantly, and for me, that is really tiring, at least after a while” (Informant 2)*


High-level attentiveness to employees through listening, observing verbal and nonverbal cues, tuning in to their problems, and perceiving the need to always show up as available, present, and committed leaders contributed to establishing a closer manager–employee relationship. However, the downside to this was that these caring actions and pressure for availability can be emotionally draining [59] and the first-line managers needed to extend their work hours to conform to these demands.

### 3.2. Practical Demands

Practical demands correspond to those demands that require dedicated action and disciplined attention to practical adjustments, e.g., establishing thorough structure and reorganizing work, and place–time constraints, e.g., requiring more time for accomplishing tasks.

#### 3.2.1. Expectations to Experiment

The company’s senior management instructed the first-line managers to experiment with work models and find *what works*, and above all, advised the managers to prioritize caring for their employees and talking to each team member individually, thus adding another layer to managerial work. They perceived their situations as being left on their own to make sense of the contents of their new role during the pandemic and, thus, faced frustration:


*“That has been difficult because…the machine is not stopping here. We are working a lot. As a manager, having 15 people to take care of on top of the rest of the work. I tell you…that has not been easy. Now the focus has been on employees a lot and I just needed to work a little more and then reprioritize things.” (Informant 6)*


Additionally, in attempts to find out *what works*, the first-line managers experimented with different work designs, e.g., four-day work week, rotation, hybrid structure, or flexibly organizing work according to tasks and activities. The managers were instructed to assess what makes the most sense in their situation (e.g., production required in-presence attendance whereas administrative benefited from other arrangements). On top of that, the first-line managers strived to connect with their employees by engaging with them through organized interaction points, e.g., one-on-ones, virtual coffee meetups, online team meetings, and through activities such as meditation sessions, online games, gatherings in outdoor spaces, virtual cocktail parties, and by sending gift packages. To care for their employees, the managers equipped them with the necessary equipment for their home offices, coached them, provided advice on establishing routines, and encouraged them to incorporate regular breaks in the fresh air and focus on their family lives. While the managers succeeded in maintaining productivity, they needed to reorganize and extend their work hours due to their increased attention toward employees. The senior management encouraged managers to experiment; however, the first-line managers lacked guidance:


*“There are no rules. We are waiting for corporate decisions on how to work from home but right now it is: “Experiment and just see what works in your team” (Informant 7)*


Whenever possible, the first-line managers tried to revert to working from the office to reduce the need to formalize meetings and reduce the emotional burden of managing their teams from a distance. However, managers’ understanding of experimentation was not aligned with the vision of the organization, and hence the CEO of the company communicated the need for readjustment:


*“The CEO sent an email where he said that he heard rumors that managers were trying to get back to the way we worked before Corona and that he was disappointed to hear that because we have learned a lot by working from home. He would expect that we would consider our team’s situation when evaluating what is best for our employees in how we work. That was his expectation. That was quite a strong way to say: “You should not strive to go back to how the world was, instead, you should try to adapt and find what is the best solution now“. If we want to experiment with different things, we can do so. If we have good examples, then we should share them with other managers, but nobody would tell us exactly what to do” (Informant 4)*


The encouragement to experiment and care for employees added additional responsibility to managerial work. The first-line managers found that experimenting and figuring things out without much support was frustrating. Furthermore, the organization empowered the decision-making abilities of the managers. However, their authority was challenged when their experimentation did not match the vision of the organization. The ambiguity of expectations further added to the experienced uncertainty.

#### 3.2.2. Establishing Structure

Leaving the physical workplace created a barrier to meeting with people, requiring deliberate action from first-line managers to overstep it, thus the managers engaged in activities that would provide them with insight into both employee well-being and performance. Even when trust in a team is high, supervision tends to increase when the workforce transitions to working from a distance [59,60]. For example, the quick daily virtual check-ins with people compensated for the lack of physical presence. Increasing the unofficial talks were intended to mimic the natural office, *bump-ins* e.g., when coincidentally running into someone at the coffee machine or hallway. However, since removing the physical workplace, the managers needed to plan for these quick exchanges deliberately:


*“All the little things you could do in 30 s when you are in your office—just by going to somebody’s desk—that gets more formalized now. I spend more time on one-to-ones than I did before because I was often all over the place talking to people (when in office)” (Informant 3)*


The first-line managers recognized that the distance barrier required more planning and structuring of team meetings:


*“The team meetings take a little more discipline to make them effective. We have learned the importance of being disciplined, and clearer on having a more structured approach to our meetings virtually” (Informant 5)*


They perceived that they needed to follow a structure to ensure that necessary topics are covered and that each employee was involved in the meeting. Their way of involving employees was by inviting everyone to verbalize their thoughts, at the same time, tracking who had contributed to the discussion: 


*“(At meetings) I have my lists where I tick off names, so I ensure that everybody has said something. I need to empower and remember everybody. I try to let it flow naturally at meetings but with my little pin on the paper so when somebody is not saying anything, I can ask: “What do you think? Oh, and what about you?” (Informant 6)*


However, the need for more robust planning and structuring of meetings, combined with interaction from a distance through the MS Teams platform, contributed to interactions becoming less intuitive and interactive. Consequently, the planning and structuring of meetings, and having agendas and outlined discussion pointers, blocked the spontaneity and demanded more thorough preparation and thus effort on the first-line managers’ part. 

#### 3.2.3. Working Longer Hours

Online interaction with employees shifted the nature of their work, i.e., distance managing required an increased focus on people and their well-being through the screen:


*“Even though the job gets done, it takes more mental power to focus on the screen and the person’s body language. You do not need to use that much energy when you are in the same room…it is draining to work on a screen so much. I think it requires more energy to have a meeting online” (Informant 4)*


Because of the attention on caring for employee well-being, during the official work hours, the first-line managers participated in online meetings with employees, and before or after the formal work hours, they worked on tasks requiring focus time. The lack of presence required more time and effort from the first-line managers. The managers experienced the check-ins to be helpful, allowing for a quick reach of people, thus they prioritized having them. However, in the online space, the managers deliberately needed to remain connected to the rest of the team:


*“It demands something different than being a manager in the office. You need to pay attention to making sure that you are connecting with employees. If not daily, then often. I do not think it is enough to call them in for one-to-one once a week and then just leave them on their own. You need to make sure that they have a close connection. They need to know to whom to go if they have questions” (Informant 2)*


The downside was that individual check-ins lead to first-line managers partaking in continuous meetings, overscheduling, and finding themselves in front of the screen for 8 to 10 h and experiencing difficulties closing work. The managers experienced that while the meetings became more efficient and took less time (25 min as opposed to 30 and 55 as opposed to a full hour), the meeting activity increased, contributing to workdays becoming longer and more intense. Due to the workdays becoming more intense with meetings, more emails, messages, and calls, the first-line managers saw that their work hours became longer and they felt trapped in longer workdays:


*“I have had long working hours and attending meetings virtually back-to-back, not only with my people but also related to other activities within the company…so despite that we have maintained our productivity, it has been extremely tiring sitting in front of the computer, maybe 8 to 10 h a day just attending meetings. I have learned by the hard experience that just sitting in front of the computer from eight to five or eight to six…that is so damn tiring, and it doesn’t work for me any longer” (Informant 5)*


Attentiveness and connecting with people were perceived as time consuming due to the abundance of meetings it demanded, consequently requiring that first-line managers work more and reorganize their workdays to include both connecting with employees and attending to their other tasks. Access to ICT further enabled the managers to keep working despite facing tiredness.

### 3.3. Work Engagement Enablers

Work engagement enablers involve resources that motivate and maintain work engagement, for example, a supportive environment and internal capacities.

#### 3.3.1. Available Organizational Support

Early on, the senior management made it clear that jobs are safe, the organization is doing well, and people will not lose their jobs. This acted as a fundamental relief for first-line managers and removed some of the burdens from the first-line managers. In addition, the organization consistently shared news from the authorities across the organization, ensuring that the rest of the organization is informed of the external situation and ways the news lands within the organization. However, the managers perceived the communication was missing concrete rules:


*“We are a part of a fantastic company. I think that communication-wise in these times, something has been missing. It is just the overall emails coming out saying: “Now we are in this situation—do this and do that…until further notice and at your discretion.” What does that even mean? They need to set rules in stone” (Informant 1)*


To guide and inspire managers, the organization offered standardized guidance on working and managing from home and support distributed through the corporate website or communicated through emails, especially at the very beginning of the pandemic. The guidance from the organization included video training sessions and inspirational recordings representing the approach to distance management. The advice focused on setting up an ergonomically sound workstation and practical advice on ways to facilitate virtual interactions. However, after offering the initial guidance, as time progressed and the government issued new restrictions and the second lockdown (see Table 1), the organization assumed that managers could already operate effectively and thus did not follow up and align with the needs of the managers.

The human resources (HR) function announced that they were available for managers in case they would like to receive inspiration and support. However, it was up to the managers to schedule these connection points, and since their workload was already high, they did not prioritize this offer. Overall, the first-line managers found the guidance helpful at the beginning but soon after they realized that there was nothing new and that the tools offered by HR tended to be a step behind and not adequately fitted to the present needs of the managers:


*“I went in and had a look at some of the training material, and I was like: “Yeah, I do that, and yeah, that is the way”…it almost felt like there wasn’t anything new” (Informant 4)*



*“There are all sorts of digital tools, which we can use. Do I use them? No, I do not. It is not that there’s no support, there is support, but it is all digital and that is not what I’m looking for these days” (Informant 5)*


The organization communicated that jobs within the company are safe thus allowing the managers to experience relief about job security within their teams. Furthermore, the company shared the decisions made by the government and explained how the government’s decisions would affect the company. However, while the tools, advice, and support were available, the managers perceived that the support was more helpful in the beginning stages of the pandemic. The managers found that the support was not timely and was lacking behind managers’ experiments and overall was not fitted to their needs and schedules. 

#### 3.3.2. Internal Capacities

While the organization offered support to some degree, the first-line managers perceived that the support was minimal. The managers expressed that they wished for more support, and they also tended to act in a self-reliant way. They saw *figuring things out* simply as a part of the managerial task:


*Interviewer: How is the organization supporting you as a distance manager?*



*Informant 5: No further comments on this.*



*Interviewer: You have recently started in a new role—has there been attention to helping you?*



*Informant 5: No.*



*Interviewer: Okay, is it just: “Here, you’re put in this position and then sort it out”?*



*Informant 5: Exactly…I guess that also comes with seniority. To be honest, I would also be a bit offended if some people would come and told me: “You have to do this and do that”. I have a feeling for what I want to do, but otherwise—there have been no special support programs” (Informant 5)*


The ability to rely on internal capacities in deciding on a direction was grounded in the seniority and experience that comes with dealing with different situations over a longer time. The reliance on own capacities was further boosted by the demand to experiment with work models:


*“The structure (on how to manage from a distance) is developing, as I’m learning to work in this way, where I find what works, and what doesn’t, and then try to increase on the things that work well. I think it could be good to have the best practices for leading from a distance that would be a big benefit for everybody. Everybody, of course, learns over time, but it’s also a lot of frustration” (Informant 4)*


As a result, relying on experience and personal abilities to figure out the next steps was another resource the first-line managers used in distance management during the pandemic. This also included them searching for guidance from LinkedIn and YouTube, by embracing flexibility, accommodating employee preferences, and above all, caring for employee well-being. 

#### 3.3.3. Supportive Socialization

For support during the pandemic, the managers mainly turned to their manager colleagues, employees, and the network within the company. With their manager colleagues, they discussed good management practices, discussed struggles, and learnings among others who were placed in a similar position. They used manager gatherings for aligning work design, gaining inspiration, and emotionally supporting each other:


*“At management meetings, we discuss aligning with other managers. I think we managers are good at communicating—we are supporting that now we are experimenting, using the opportunity, and figuring out what is working (Informant 7)*


The first-line managers realized that both when managing from the office and distance, interactions, discussions, and building on ideas with employees and peers and overall, a positive team spirit tended to drive and keep the motivation and work engagement. The managers reflected that throughout the COVID-19 pandemic, they realized the significance of social life at work:


*“For many of us…it has also become very clear that the social life at work is extremely important. I think many of us have realized that it is extremely important to have colleagues around. It is not rocket science what I’m saying here but just some basic observations” (Informant 5)*


The interactions and connections became especially significant when working from home. Knowing the way around the company and having a network across the organization as well as connecting and perceiving a bond with employees provided a sense of well-being to managers, which catered to their self-assurance making them more secure. This is because social support is an influential resource that alleviates the effects of demands by helping individuals to cope better with them [42]. For example, when connections between the manager and their employees were filled with a positive attitude, they alleviated the experienced distress and improved collaboration over distance:


*“One of the benefits I have harvested is that my team likes to spend time together and be social. There is a very open atmosphere, and they even do things after work. I think that has helped in these times to keep up a good atmosphere, and everybody’s trying to have a positive mindset, even though they are kind of tired of the situation” (Informant 4)*


The first-line managers appreciated the commitment and enthusiasm demonstrated by employees for engaging and striving to remain connected to each other in the team. The enthusiasm spread by the employees also affected the managers, hence, creating a positive feedback loop where satisfied employees influenced the sense of well-being in managers too, despite the circumstances. The first-line managers experienced a sense of well-being from the socialization with their teams. Social support such as team cohesion, and a supportive work environment in general, is a powerful resource that facilitates well-being and achieves work goals [13]. 

## 4. Discussion

This study aimed to understand first-line managers’ perceptions of their job demands and resources available to them when transitioning to distance management during the first year of the pandemic. The study explored the particularities that influenced extending their work, hence the study responds to the calls by Ipsen et al. [19] and Hassard and Morris [30].

In line with other studies (e.g., [19,32]), we found that the first-line managers perceived the transition to distance management as challenging, where they were forced into distance management without proper training [59], which led to longer workdays [61]. More specifically, we observed that the transition to distance management shifted the job contents for first-line management. The job demands the managers faced included emotional, e.g., the need to handle uncertainty when managing employees from a distance during the pandemic and increased focus on prioritizing care for employees. 

Additionally, the managers faced practical job demands, e.g., the need to respond to the organizations’ expectations for experimentation by placing additional effort on structuring meetings and reorganizing work. As a response to work becoming more demanding, the first-line managers worked more (e.g., between 8 and 11 h) and the managers could remain connected to their work through ICT. The first-line managers devoted their official work hours to back-to-back meetings, whereas mornings and evenings, which they used for commuting to the workplace before the pandemic, were now dedicated to tasks requiring concentration. While the first-line managers received support from the organization to a higher or lesser degree, they relied on work engagement enablers, e.g., obtaining support from their employees, managerial peers, and network within the company, and their internal capacities, e.g., experience and resourcefulness to solve issues and motivate work engagement. 

### 4.1. Emotional Demands Accompanying Transitions

The sudden transition to working from home was the main reason the first-line managers perceived leadership under the pandemic as highly challenging. The first-line managers were continuously adjusting to the changing COVID-19 states that moved the workforce from the office to working from home and back. This required the first-line managers to work as usual, leading employees from a distance despite missing direction and lack of training in their new management task. The managers were not and could not have been prepared to face such a long and complex transition that required them to navigate between onsite and offsite, predominantly managing their employees from a distance. The first-line managers needed to respond to ambiguous tasks, for example, making immediate decisions for employees (e.g., deciding who stays home when), simultaneously, they were not always aware of the official guidance and company policies. 

When leading employees during the pandemic, all the first-line managers shifted to prioritizing a caring relationship with their employees, relying on care to maintain employee well-being and performance. When managers led their employees from a distance, they discovered the importance of visibility in their daily management. Through seeing employees regularly at the office, the first-line managers perceived that they knew how employees were dealing with tasks, and thus, transitioning to distance management made them realize the extent to which they relied on daily visibility to detect cues about employee well-being and performance. Because of the lack of daily visibility, we observed that the first-line managers experienced an underlying concern for their employees. Therefore, they assumed that caring for employees was necessary. The senior management further stressed prioritizing caring for employees. This added another feature to the first-line managers’ work. As a result, the primary task of first-line management, i.e., performance-oriented supervision of employees [39], turned to people-oriented leadership even more than before the pandemic. By supporting employees, empowering them, facilitating their growth, and sheltering them, the first-line managers adopted the characteristics of a caring leader, i.e., a leader who strongly cares for employees [62]. A leader who is considered to be caring demonstrates personalized attention to individual employee needs and is present, visible, accessible, and devoted to employees [63]. At the core of such caring are the leaders’ presence and availability [64]. In line with the caring leader traits, the first-line managers expressed that their caring behaviors included demonstrating interest, attuning to employees and their life events, listening and being there for them, making employees feel seen and understood, and cared for, and offering a prolonged availability. On the one hand, such caring enhanced closeness in the manager–employee relationship. Showing care to another tends to cultivate a relational connection that may improve employee engagement and bring greater employee commitment and performance [65]. On the other hand, the challenges of caring leaders are widely relevant [62] as caring actions, and pressure for availability can be emotionally draining [59]. In practice, caring can be demanding and time consuming, and overall, an impossible ideal that poses practical limitations, e.g., providing extra care limits time and energy devoted to other areas. Leaders are expected to care for employees [62,66,67]. However, the expectation of leaders to care in all possible ways may lock them in unhealthy and exhausting relationships [63]. Furthermore, the expectation from first-line management to turn into caring leaders might be an unreasonably elevated demand as first-line management is expected to direct employees and monitor how work is accomplished [39]. However, this demonstrates how responsibilities from management levels above tend to pass on to first-line management and how first-line management embraces greater responsibilities [39].

### 4.2. Practical Demands in Distance Management

When shifting to working from home, the organization encouraged the managers to experiment with *what works* and care for their employees as an approach to their daily management, hence imposing additional demands on managers. Experimenting involved the first-line managers finding a framework that would fit adequately to their teams’ situation, depending on the operational area. Consequently, the managers tried different work designs, e.g., a four-day workweek, rotation, hybrid work structure, or organizing work flexibly according to tasks and activities, sticking to things that work, and developing their approach as time proceeded. However, the first-line managers were experienced in onsite management. The managers tried returning to managing from the office whenever possible so they could preserve their resources and manage employees in the way they felt most competent. On the surface, the first-line managers were empowered by the organization to experiment in discovering the most suitable ways of working and deciding for their teams. However, when the CEO of the company tried steering the managers back to distance management by expressing disappointment that managers and their teams were returning to a pre-COVID-19 way of operation, it demonstrated that the empowerment was handed with a covert expectation to embrace working from a distance, despite the effectiveness. Nevertheless, the first-line managers were expected to deliver organizational goals as usual. This points towards the challenge of lower-level management that despite being responsible for the performance of their teams, they have a low influence in decision making and control in their roles [39].

First-line management is responsible for managing daily work operations, focusing on their team, and reacting to situations at hand [39]. Hence, their daily presence at work is important [34]. We observed that the first-line managers strived to be present, available, and visible daily through ICT. However, the managers believed that the lack of presence and the need for more discipline and structure in interactions required more preparation, contributing to the managerial tasks requiring more vigorous work effort and input than in the office. Since employees were out of sight of the managers, the managers perceived that to ensure efficient interactions and meetings, they needed to prepare more and have a structured approach. This was because connecting to others through ICT created a barrier, making people’s work behaviors invisible and, as a result, altering managerial oversight. Distance challenged the managers and as a result, they tended to increase interactions to bring employees closer. However, when connecting to employees at a distance, there is limited time for influencing individuals and extracting information [60]. Therefore, in attempts to cover all critical issues and include everyone at meetings, the first-line managers created and followed agendas, ticked-off names of those who had contributed to a meeting, and pointed toward those who had not. The first-line managers acknowledged difficulties in trying to mimic onsite interactions and instead perceived distance leading to rigid, less intuitive, and spontaneous, and overall, more effortful and time-consuming meetings. Yet, engaging in interactions and applying a more structured approach allowed managers to gauge their employees’ well-being and performance. The increase in meetings and interactions functioned as an antidote to not seeing employees. However, perceiving the need to continuously interact and be available online also connects with the *always-on* culture, thus blurring the boundaries between private and work life and causing a lack of balance between these domains [68]. Idealistically, when working from home, the managers could control their job content, tasks, and time, and voluntarily engage and disengage from work. For example, studies pre-COVID-19 suggested that working from home improves well-being and work–life balance [69,70]. However, we observed that access to ICT, large volumes of work, and the work environment at home kept the first-line managers connected to their work. This can result in difficulties in distinguishing the beginning from the end of their work [6,25,71,72], and blurring of professional and private lives [30], which we also observed in our study. 

### 4.3. Facing Scarce Organizational Support 

The case company is known for providing and valuing high well-being and proper working conditions for their employees (according to the company’s internal communication and confirmed through www.glassdoor.dk). While first-line managers are also employees within the company, through this study, we found the company did not treat the first-line management’s well-being as a priority. The organization provided support for the first-line managers, e.g., sharing information from the authorities and standardized guidance and advice on working and managing from home, yet organization-wide and coordinated support from the human resource function and senior management was lacking. Human resources announced availability in case the managers needed inspiration and support. However, while the tools, advice, and basic support were available, and managers found the guidance helpful to some degree (especially during the first lockdown), simultaneously, they perceived that the support was not timely and not what they were looking for. Even though human resource managers have a significant role in supporting workers during an event such as COVID-19 [73], we observed that the human resources were not attentive to first-line managers’ needs. Overall, the influence and effects of the transition to a new management task during the pandemic were not considered. This is potentially because human resources were not sufficiently equipped for such a large transition having no prior experience to draw on, and the senior management might have held on to the hope that the pandemic would soon pass. 

When the organization required that the managers experiment and care for their employees, they remained unaware of the high emotional demands that they placed on the first-line management. The demands to experiment and care were profound by themselves because of the uncertainty the first-line managers experienced from the pandemic and the minimal support available from the organization. 

### 4.4. Work Engagement Enablers as the Means to Well-Being at Work

Overall, the transition was demanding and experiencing a lack of support from the organization, the first-line managers turned to relying on work engagement enablers (e.g., social support, experience, and resourcefulness) to guide their actions further. The reliance on internal capacities was especially triggered when the first-line managers were instructed to experiment. Furthermore, due to the lack of physical presence and daily interactions at the office, the significance of social life at work became especially notable during the pandemic. Having connections in the company, the feeling of belonging, and being a part of a group are necessary for maintaining well-being at work. 

Even though the first-line managers experienced high demands, our study demonstrates that relying on work engagement enablers such as social support kept them motivated and engaged. Social support is among the most effective resources when coping with high demands [42], and studies (e.g., [34,74]) have observed the positive effect social support has on well-being. For example, first-line managers with proper social support at work tend to be less affected by burnout symptoms [34]. We observed that the first-line managers turned to their employees, manager colleagues, and networks within the company for support. The social support and the supportive environment provided the managers with space for sharing and experiencing solidarity during the pandemic challenges. The first-line managers found that interactions and connection with employees and colleagues boosted their energy and overall reduced the tiredness and exhaustion they experienced from working from home. Especially, a positive team spirit promoted first-line managers’ motivation and work engagement causing a feedback loop where satisfied employees triggered a sense of well-being in managers too. The social environment alleviated the experienced distress, proving to be a powerful resource that facilitates well-being, enhances motivation, and thus is important when achieving work goals [13]. This confirms the importance of having social support with high pressures during challenging times.

### 4.5. Implications for Managerial Well-Being 

First-line managers’ workload during COVID-19 increased, prolonging their work hours, especially affecting lower-level managers as they were pulled into attending to employees while needing to remain on track with delivering organizational goals [32]. Managers tend to deal with pressures to accomplish more with less while facing continuous change [75]. Simultaneously managers’ well-being is rarely addressed while managers deal with their issues in isolation or within their immediate social circles [13]. According to the job demands and resources model, when the job demands are higher than the available resources, an individual may develop burnout; thus, having access to job resources is especially significant when job demands are high [43]. The access and availability of resources alleviate the impact of the high job demands.

Our findings demonstrate that the first-line managers perceived the transition to distance management as stressful, especially, the need to deal with uncertainty during the pandemic and prioritize care for employee well-being required managers to dedicate emotional input. The cost of such intense caring and continuous listening to distress may lead to experiencing depletion and fatigue. Furthermore, the ties to work through ICT affected managers’ work–life balance and, over time, extended work hours and screen time, resulting in first-line managers experiencing loss of energy and depletion from the high demands they faced. In addition, the inability to detach from work prevents individuals from properly recovering and further depletes their resources [59]. 

We observed that the managers experienced tiredness from their pressing demands and tended to develop negative attitudes, i.e., cynicism towards distance management and the available support provided by the organization. The first-line managers wanted to be among people, and they perceived that support was lacking while they were struggling with their additional workload. Cynicism and tiredness are potential signs of developing burnout, thus raising a concern that the high pressures may result in managers developing a burnout [50]. Therefore, future research needs to explore managerial well-being further and seek to understand how changes in managerial job contents affect their job quality.

### 4.6. Theoretical Contribution

The major contribution of this study is the exploration of first-line managers’ experience transitioning to distance management, thus contributing knowledge to the emerging research of work under the pandemic.

Furthermore, previous research rarely differentiates the different managerial levels [34]. Consequently, our study contributes to the managerial well-being literature by identifying the constituents of emotional and practical job demands and work engagement enablers and their influence on managers’ job quality in the transition to distance management. Moreover, based on the results of the study, we identified how managers faced scarce support when undergoing the transition. As a result, we suggest dedicated actions organizations can take to support managers throughout a transition.

### 4.7. Practical Contribution

Managers’ leadership behaviors influence employee well-being as well as loyalty toward their organization [14]. Instead, the lack of organizational support may significantly affect managers and employees. Thus, it is important to be attentive toward managers’ well-being and focus on improving their conditions for leadership and work conditions on a regular basis [12,13]. Organizations must become aware of the need for providing organizational support, including in-house support such as human resource management [56] and support focusing on individual managers.

Actions focused on supporting managers could include monitoring managers’ workload and well-being to understand the impact and implications over time. These actions would include the organizations actively involving managers in the identification of their job demands and influential resources and their effects on their well-being. Monitoring would allow organizations to continuously optimize managers’ job contents and provide timely support according to the needs of managers and equip them with necessary resources. The support could include interventions focusing on job redesign and task restructuring and support towards role-related practices and relaxation techniques [50]. In addition, organizations could focus on establishing knowledge-sharing networks and providing a supportive social environment from which the managers can learn, develop skills, and practice healthy management practices [13,50]. Facilitation of these networks would act as support in undergoing transitions and prevent the development of burnout [37] and increase managers’ work engagement. Furthermore, organizations could develop, test, and refine new leadership principles and systematically implement them in the whole organization while monitoring and continuously improving the principles. Furthermore, especially if workplaces adopt more remote work and distance management post-COVID-19, managers working from a distance must acquire new skills and accept the change in their jobs. For this, organizations can assist with knowledge informing the development of guidelines on how to support those managing from a distance.

### 4.8. Recommendations for Management

From our findings, we can derive recommendations for senior and first-line management on improving remote work, which has become the norm in many organizations following the pandemic.

First, we outline the recommendations to senior management on how they can improve conditions for first-line management managing from a distance, and after that, we elaborate on operational approaches first-line management can undertake to improve remote work.

#### 4.8.1. Recommendations to Senior Management on How to Improve Conditions for First-Line Management Managing from a Distance

Based on our study, we have identified significant directions that senior managers can follow. To be specific, we recommend that the senior management prioritize the well-being of first-line management. Such prioritizing would include proactively reaching out, demonstrating interest, and showing appreciation to first-line management. Furthermore, it is important that first-line management becomes involved in decision-making processes and senior management permits more control to them in their roles, thus allowing the first-line managers to become an integral part of organizational changes. If first-line management is accountable for the performance of their teams, they need to be able to participate in decision making. Simultaneously, the senior management must empower and allow first-line management to experiment and trust in their decisions, follow up on progress, and provide relevant support along the way. 

Additionally, the senior management needs to place attention on providing official guidance, setting clear rules on how to work remotely, as well as ensuring and providing models and examples for remote work. It is also important that they set clear expectations, and align them with the first-line management to reduce uncertainty for first-line management undertaking the distance management task. By mapping out the necessary resources and striving to equip first-line management with these resources, senior management can practically assist first-line management in their distance management task. Furthermore, if senior management would establish ongoing training for distance management and virtual psychology, that would allow the first-line managers to deal with distance management responsibilities better and provide an understanding of how to decode people’s emotional states online. Gaining such an understanding would allow first-line managers to understand their employees better and, as a result, become better able to prevent negative outcomes.

Regular in-presence face-to-face and online interactions bridging all management levels is important for knowledge exchange and socializing. Hence, we recommend that the senior management supports and facilitates peer network establishments involving all managerial levels and creates a dedicated space for socializing, learning, sharing, and receiving emotional and practical support. Engaging in in-presence face-to-face contact regularly (biweekly or as agreed between the parties) is important for both first-line management, senior management as well as employees as it allows connecting, reading people, and keeping the people together. Furthermore, setting up regular (biweekly or as agreed between the parties) online interaction points, e.g., one-on-ones and online network meetings would allow for follow-up on how the first-line management is coping with their immediate demands, discuss workload, and provide the necessary support.

#### 4.8.2. Recommendations to First-Line Management on How to Improve Remote Work

Informed by our findings, we recommend that the first-line management connects and collaborates with senior management. This would include first-line management sharing their struggles and reaching out for support to senior management so senior management is aware of the blind spots and can provide the support first-line managers need.

On a team level, to ensure successful remote work, first-line managers need to align their expectations with the team. Achieving such alignment would include setting clear expectations on what needs to be delivered by the team members, and when, and organize a follow-up. When working remotely, it is important to establish common agreements and rules among the team on how to conduct online meetings and set rules for online meetings, for example, requesting that all participants engage in remote meetings through individual devices even though they are physically present at the workplace, requesting to switch on the camera when participating in online meetings, and setting 25 and 55-min long meetings, thus allowing a 5-min break between each meeting. In addition, creating agreements to strive to handle conflict situations and clarify misunderstandings in a face-to-face setting or at online meetings with a camera switched on. For conducting productive online meetings, it would be fruitful to assign a dedicated meeting facilitator for planning, preparing, and structuring.

Furthermore, we acknowledge that when working remotely, maintaining team cohesion is important. This way, first-line managers can cultivate team cohesion is by agreeing with the team on interaction points, and interacting regularly. Examples of specific interaction points would include having a short online check-in three times a week for quick updates, problem-shooting, and socializing. In addition, organizing a biweekly online one-to-one meeting with each employee to discuss performance, workload, and his or her well-being. Additionally, opportunities for employees to socialize and support each other should be encouraged and facilitated. Furthermore, scheduling frequent in presence face-to-face encounters with the team (e.g., on a weekly/biweekly basis) for socializing and keeping the team spirit. Regular face-to-face interactions allow first-line management to observe and decode employee behaviors and take action. Brainstorming and other interactive team activities may be conducted in the office for better results. Because a good social environment is among the top resources in facilitating dealing with challenges, the first-line management needs to participate actively in manager networks to share and learn from manager peers.

To ensure their own well-being, first-line managers need to strive for reaching a balance between people management and their other tasks. Practically, achieving such balance would require that first-line managers craft their work day by setting up a dedicated time for interacting with employees but also for concentration work and strictly block this time in their calendar. Furthermore, they need to set boundaries for on and off work and prioritize disconnecting from work. During work hours, taking regular breaks, e.g., every 30 min stepping away from the screen, going for a walk, and moving frequently during the day would allow for reducing depletion and fatigue from the extensive screen time. Lastly, we would encourage the development of dedicated rituals for closing work, e.g., closing the computer at a certain time every day and going for a walk or engaging in another activity that would allow shifting from work to life. 

Engaging in remote work and distance management offers flexibility but it also requires reshaping of practices, new structures, and processes in order to ensure optimal well-being and performance of remote workers. Additionally, these practices, structures, and processes need to be implemented immediately when transitioning to remote work and distance management in order to prevent the rooting of obstructive habits that may eventually lead to negative organizational outcomes. Once the new practices, structures, and processes become an integral part of the organizational standard, they can be relaxed as needed and when agreed upon by the different parties within organizations.

## 5. Limitations

The strength of this study is that it involves the empirical data that were collected throughout the first year of the pandemic, thus adding to the previously accumulated knowledge on distance management and remote working. However, the data represent the unique demands and resources during the transition to distance management during the pandemic. Transitioning to distance management outside the COVID-19 context would potentially present different demands and resources. Furthermore, our study focuses on a single case, exploring the perceptions of seven first-line managers. The results gathered from a narrow sample require retrospectively exploring the perceptions of managers in other organizations to understand the job demands and resources during COVID-19. Including a larger pool of managers would also prove beneficial to observe whether managers in other organizations perceive the transition differently than the informants present in this study. Compared to other European countries, the Danish workforce had already gained some experience in working from home and distance management before COVID-19. Therefore, the demands and resources for managers in another country’s context might appear different.

Furthermore, we have relied on managers’ accounts of their demands on resources. We do not have insights from the employees. Employee perspectives would have informed their perceptions of how managers dealt with uncertainties, experimentation, conducting structured meetings, and employee perceptions on managerial caring, and whether their point of view is aligned with the managers’ outlook. In addition, we did not measure the objective change in managers’ work hours. Additionally, organizational support was not measured objectively. Instead, we relied on managers’ subjective accounts.

Therefore, we propose that more studies explore managerial job demands and resources in distance management outside the context of the pandemic to understand whether COVID-19 affected managerial work contents permanently. This would provide an opportunity to validate the results of this study.

## 6. Conclusions

The COVID-19 pandemic affected workplaces across the world and introduced managers and organizations to distance management with no collective experience to draw on. Because of the significant influence of the pandemic and that previously accumulated knowledge on remote work could not apply to the COVID-19 context [2,31], remote work and distance management during COVID-19 is still an emerging discussion.

This qualitative study explores seven first-line managers’ perceptions of their job demands and available resources while transitioning to distance management, from May 2020 to May 2021, and the study indicates that the change in job duties might have negative implications on first-line managers’ well-being. While previous studies have explored employee job demands and resources, our study focuses explicitly on first-line management, which is a vulnerable group that is prone to experiencing stress at work due to them facing complex, conflicting demands, and simultaneously having a low influence in decision making and control in their roles [34,35,38]. Early pandemic studies point out that managers experienced the transition to distance management during the pandemic as highly challenging [19,20]^.^

The significant contribution of this study is that we provide insights on the particular perceived job demands and resources by the first-line managers, thus taking a step forward by explaining the particularities that influenced the extensification of first-line management’s work. Our findings suggest that the first-line managers faced new job demands. We observed that the first-line managers experienced emotional demands, e.g., facing uncertainty when managing employees from a distance and increased focus on prioritizing care for employees. We also found that managers increased practical demands, e.g., the obligation to experiment with different work designs and activities, which required first-line managers to put additional effort into structuring meetings and reorganizing work. Consequentially, to fulfill these demands, the first-line managers needed to extend their work hours. Overall, the first-line managers experienced the forced transition to distance management as stressful. Especially, the need to care for employee well-being required managers to dedicate profound emotional input. Concurrently, they received minimal support from the organization; instead, they relied on work engagement enablers such as supportive socialization with employees, colleagues, and manager networks, and their internal capacities. The voluminous workload and demands affected their well-being, contributing to experiencing pronounced tiredness from managing from a distance and triggering the development of cynical attitudes toward the support offered by the company. Furthermore, we observed that access to ICT and the work environment when working from home kept the first-line managers connected to their work, thus affecting first-line managers’ work–life balance. First-line managers perceived that it was difficult to follow the activity of employees, and while the increase in meetings and interactions functioned as an antidote to not seeing employees, it set the first-line managers in an *always-on* mode, keeping them connected to work through ICTs and challenging their disconnect from work. Over time, the extended work hours and screen time resulted in managers experiencing a loss of energy and the inability to detach from work, further depleting their resources [59]. Our findings indicate that if the first-line managers do not eventually receive more support or have the possibility to recover their resources, they might face eventual burnout.

Even though the company in this study emphasizes that they value employee well-being, we observed that the first-line managers’ well-being was considered lightly during the pandemic. As the forced transition to remote work and working remotely in a collective way was new both in Denmark and abroad, we may assume that managers in other companies experienced the transition similarly. However, further studies with a retrospective approach would be needed to confirm this assumption.

Furthermore, in this study, we observed that the organization approached the shift to remote work and distance management as a geographically transferable task, i.e., changing the location of where work is conducted but not aligning the practices and structure. Hence, the organization did not see it as an organizational change. In remote work, people do not work in the same place, at regular work hours, and within the same work culture; therefore, remote work requires the development of new practices and structures [29]. The lack of awareness about the significance of the transition contributed to the organization adding layers to first-line management work (e.g., to care for people and experiment with work designs), yet leaving these managers to deal with their demands on their own while having minimal resources. As a result, the first-line management had to settle with whatever tools were available at hand [76], improvising with new approaches. The non-existent feedback loop between the organization and the first-line managers fostered ignorance about first-line management’s workload and struggles.

The findings from this study provide new insights to organizations about introducing a transition and establishing awareness for providing organizational support to first-line managers. The support needs to focus on monitoring managers’ well-being while continuously collaborating with first-line managers to optimize their job demands, identifying the most influential resources, and actively seeking to equip managers with them. Furthermore, observing that responsibilities from management levels above tend to pass on to first-line management and how first-line management embraces these greater responsibilities, we believe future studies need to further explore ways to support the vulnerable position of first-line managers during transitions. Additionally, the development of new practices and structures in the transition to remote work remains a highly relevant subject for future studies.

As a final remark, it is important to mention that this study was conducted during the COVID-19 pandemic; therefore, further studies need to explore managerial job demands and resources in a post-COVID-19 setting. These studies need to observe the degree to which managers face increased workload, whether their job demands have changed post-COVID-19, and explore the effects of the managerial job contents on their well-being after the pandemic.

## Figures and Tables

**Table 1 ijerph-20-00069-t001:** Informant participation in interviews connected to the chronological COVID-19 developments in Danish society and workplaces.

Round Number	Round 1	Round 2	Round 3	Round 4	Round 5	Round 6	Round 7	Round 8	Round 9	Round 10
Timeline	May 2020	June 2020	Aug 2020	Sept 2020	Oct/ Nov 2020	Nov/Dec 2020	Jan 2021	Feb 2021	April 2021	May 2021
States	First opening after the lockdown		Lifted restrictions	New restrictions	Second lockdown				Second opening/ re-opening	
Work mode	Engaging in partially working from home and office		Return to office	Return to working from home	Advised to work from home				Gradual return to office	
Informant 1	X	X	X	X						
Informant 2	X	X		X						
Informant 3	X	X	X	X	X	X	X	X	X	X
Informant 4	X	X	X	X	X	X	X	X	X	
Informant 5	X	X	X	X	X	X	X	X	X	X
Informant 6	X		X	X						
Informant 7	X	X	X	X	X	X	X	X	X	X

**Table 2 ijerph-20-00069-t002:** The overview of the informants’ profile (May 2020).

	Position	Management Experience	Distance Management Experience	Distance Management Span	Direct Reports
Informant 1	Global director	2.5 years	Since COVID-19 started		9
Informant 2	Global director	8	8 years	2 employees in Bangalore (1 year) and 1 in France (8 years)	22
Informant 3	Senior director	16	2 years	2 employees in the United States and 2 in India	14
Informant 4	Department manager	8	3 years	1 employee in India	11
Informant 5	Senior director	20	5 years	Virtual team, leading employees from multiple locations, e.g., Germany and the United States	20
Informant 6	Director	5	Since COVID-19 started		15
Informant 7	Department manager	15	10 years	A couple (neitherthe exact number nor location are specified)	10

## Data Availability

The data presented in this study involve interview transcripts and due to the confidentiality of the informants, it cannot be shared.

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
