# Peer review of "Exploring Managerial Job Demands and Resources in Transition to Distance Management: A Qualitative Danish Case Study"

_ijerph, 2022, doi:10.3390/ijerph20010069_

Round 1
Reviewer 1 Report
Thank you for the timely work, interesting approaches to solving current problems. Indeed, the question of transforming the usual form of off-line work into on-line format requires separate research. And recent studies reflect that simultaneously with the fall in production in 2020 due to the pandemic, the volume of information services and production of computer equipment has increased (https://doi.org/10.3390/economies10110278). This indicates a significant increase in remote work. Accordingly, the work format of most workers in organizations, including managers, has been transformed.
The manuscript is quite clear about the research methods and results processing, the necessary research tools are provided. Much attention is paid to the discussion of the identified problems.
The work reflects the primary results of the interviews, which are grouped according to common features (Appendix C).
At the same time I will highlight the following shortcomings.
1. the study was conducted on the basis of only one company in the pharmaceutical industry. The paper needs to make a conclusion about the possibility of extending the results to the management personnel of other companies. One could, of course, simply point out the limitations of the study, based on a very narrow approach to the criterion of respondent coverage. From my experience, I know that opinions about remote work are often contradictory. Many executives see great benefits in working remotely, both in terms of its efficiency and the organization of their time.
2. Only 7 people participated in the study, and only 5 of them made it through all phases of the study. The authors need to provide evidence that this is sufficient to form representative conclusions.
3. Recommendations to senior management and first line managers to improve telecommuting should be more clearly stated. I believe that the authors' study should provide for this.
I wish you the best of luck in your future work!
Author Response
Thank you for your constructive feedback on our manuscript and your helpful input to improve the manuscript. We enclose our detailed replies to reviewer comments.

Reviewer 2 Report
I appreciate the opportunity to review this article, which it aims to explore first-line managers’ perceptions of job demands and available resources during the first year of the pandemic and understand the implications for first-line managers’ well-being. The literature review presents a sufficient number of articles and authors consulted and covers topics. The paper highlights managers' leadership behaviors influence employee well-being as well as loyalty toward their organization. The methodology allows reproducing the research as it details how the interviews and periods were carried out, in addition to presenting the research instruments in the appendices. The results and analyzes present a faithful view of the answers of the 7 interviewed managers and allow the elaboration of the conclusion.
Author Response

(The authors gave the same response as above.)

Reviewer 3 Report
Dear authors
It was my pleasure to review your manuscript entitled “Exploring managerial job demands and resources in transition to distance management: a qualitative study ” and advise you to prosper your current research project. In my view, your topic has touched on a critical issue in a fascinating context. However, there are many spaces to be improved in terms of argumentation, theoretical background, research method, and findings. I hope my below comments would help you develop your work into groundbreaking research in your domain.
Introduction
A concise introduction to enable the reader's understanding of the research problem.
• Introduce the paper describing what the paper is about. Expand to emphasize the problem leading to a clear set of research questions addresses.
• Give readers a one-line preview of the other sections of the paper.
The positioning of the paper is not entirely clear. The author is better to explain the gap in this article further.
Theoretical literature has not been considered and reviewed. It’s better to observe the connection between the contents. Try to explain everything except the topics in order to establish the necessary
coherence.
The method should be adequately described to show how the research was conducted to improve clarity and transparency. This is one of the most critical parts of the paper.
The number of tables used in the article is large, the authors should recheck and correct them.
What are the theoretical and practical implications of your study and which limitations and possible future research emerge from it? At the moment. the chapter is that is now entitled as "Conclusion" should link back to the literature and show theoretical contributions, that exceed the conclusion that some literature was "inline" with the findings of the authors.
The authors need to draw substantive conclusions from their results, and suggest, develop recommendations for further research.
- Using the following reference could be beneficial as these add more evidence to the literature review section:
Measuring the Impact of Simulation-Based Teaching on Entrepreneurial Skills of the MBA/DBA Students In Technology and Entrepreneurship Education (pp. 77-104). Palgrave Macmillan.
Best of luck with the further development of the paper.
Author Response

(The authors gave the same response as above.)

Reviewer 4 Report
In my opinion, the paper is well-written, including the presentation of materials and methods, results, discussion, theoretical and practical contribution, limitations, and finally, conclusions.
I have only one suggestion. In the abstract, the authors have written “The study draws on 49 semi-structured interviews with seven first-line managers from a large pharmaceutical company in Denmark, …”. I suggest rephrasing the subtitle of the paper towards “a Danish case study” to indicate that results were obtained from the Danish experiences.
Author Response

(The authors gave the same response as above.)

Round 2
Reviewer 1 Report
The manuscript has been greatly improved and comments and suggestions have been taken into account. Some problems in the rationale for the results are described by the authors in the limitations section.
I wish the authors success in further research on the transformation of governance processes in a pandemic.
Reviewer 3 Report
Dear author(s)
Hope you are doing well. According to the review of this article, the corrections have been made.
Good luck